# Communications and High-Precision Positioning (CHP2): Hardware Architecture, Implementation, and Validation

**DOI:** 10.3390/s23031343

**Published:** 2023-01-25

**Authors:** Hanguang Yu, Andrew Herschfelt, Shunyao Wu, Sharanya Srinivas, Yang Li, Nunzio Sciammetta, Leslie Smith, Klaus Rueger, Hyunseok Lee, Chaitali Chakrabarti, Daniel W. Bliss

**Affiliations:** Center for Wireless Information Systems and Computational Architectures (WISCA), Arizona State University, Tempe, AZ 85281, USA

**Keywords:** distributed coherence, two-way ranging, unmanned aerial vehicles (UAVs), alternative positioning, navigation, and timing (APNT), urban air mobility (UAM), communications, navigation, and surveillance (CNS), autonomous vehicles, spectral convergence

## Abstract

Spectral congestion and modern consumer applications motivate radio technologies that efficiently cooperate with nearby users and provide several services simultaneously. We designed and implemented a joint positioning-communications system that simultaneously enables network communications, timing synchronization, and localization to a variety of airborne and ground-based platforms. This Communications and High-Precision Positioning (CHP2) system simultaneously performs communications and precise ranging (<10 cm) with a narrow band waveform (10 MHz) at a carrier frequency of 915 MHz (US ISM) or 783 MHz (EU Licensed). The ranging capability may be extended to estimate the relative position and orientation by leveraging the spatial diversity of the multiple-input, multiple-output (MIMO) platforms. CHP2 also digitally synchronizes distributed platforms with sub-nanosecond precision without support from external systems (GNSS, GPS, etc.). This performance is enabled by leveraging precise time-of-arrival (ToA) estimation techniques, a network synchronization algorithm, and the intrinsic cooperation in the joint processing chain that executes these tasks simultaneously. In this manuscript, we describe the CHP2 system architecture, hardware implementation, and in-lab and over-the-air experimental validation.

## 1. Introduction

Positioning and communications have become critical services in both the public and private sectors, especially with the advent of personal UAV [1,2], self-driving vehicles [3], and UAM [4,5]. These services enable critical infrastructure applications such as collision avoidance [6], traffic management [7], automated navigation [8], and V2V communications [9]. Legacy approaches provide these services using separate radio systems, which increases spectral congestion and introduces mutual interference [10]. This approach has provided an immediate solution that successfully enables numerous applications, but recent results in the field of RF Convergence [11] indicate that co-designed radio systems may achieve better performance, especially in increasingly congested environments.

We developed the CHP2 system [12] to simultaneously provide communications, time synchronization, and relative positioning services from a single radio platform. CHP2 executes these tasks using a single, co-use waveform, which limits SWaP consumption and spectral utilization. By leveraging modern co-design techniques, this system consistently achieves ranging precision below 10 cm on each transmit-receive antenna pair in a single 100 ms round-trip exchange. In controlled laboratory environments, this precision has been driven as low as 5 mm with only 10 MHz bandwidth while simultaneously enabling network communications and digitally synchronizing distributed users within a fraction of a nanosecond. When the platform geometries are known, these multi-antenna distance estimates may be translated into position and orientation estimates, the performance of which is subject to the geometric dilution of precision [13,14]. This precision enables sensitive applications such as automated landing and collision avoidance, while the small form factor allows installation into small vehicles such as UAV.

### 1.1. System Overview

The CHP2 system simultaneously enables network communications, distributed coherence, and relative positioning using a single, narrowband waveform. This waveform contains a communications payload and several reference sequences. The communications payload carries network communications packets between users and also contains shared timing information. The reference sequences estimate the ToA at each receive antenna in each frame. The ToA estimates and shared timing information drive a distributed synchronization algorithm, which estimates the ToF between each transmit-receive antenna pair, digitally synchronize the users, and estimates the relative position and orientation of the users. This system architecture is described in greater detail later in the manuscript.

The intrinsic resolution of this TWR system is proportional to the inverse of the signal bandwidth, which is approximately 30 m for a 10 MHz waveform. We leverage integration gain from the long navigation sequences to maintain high ISNR (∼45 dB) to support oversampling techniques that improve this resolution by a factor of 200 (<2 cm). This precision may be further improved by Kalman filter tracking solutions and more efficient firmware implementations. For multi-antenna platforms, these ToF estimates are extended to estimate relative position and orientation, the precision of which depends on the specific network configuration. This technology may be installed on numerous types of vehicles or stationary platforms, enabling a highly-reconfigurable network that can quickly adapt to different use cases. An example network topology is depicted in Figure 1.

### 1.2. Prior Work

The proposed work is primarily motivated by the two-way ranging system demonstrated in [15,16,17]. In these publications, the authors demonstrated a high-precision, two-way ranging prototype and explore tractable time-of-arrival estimation techniques. We extend these results by integrating modern RF convergence design principles [11], developing a complete hardware implementation on consumer-off-the-shelf hardware [18,19], and demonstrating a robust ranging prototype in over-the-air flight tests [14].

### 1.3. Motivation, Limitations, and Applications

We designed the CHP2 system to enable communications, synchronization, and positioning services from a single platform. Many existing technologies already offer these different services; notably, the GPS offers high-precision positioning and timing services, while modern cellular networks offer high-speed communications services. CHP2 does not outperform these solutions, nor do we present it as a competitor to these technologies in any capacity. Instead, we developed CHP2 as an accessible and efficient option for a subset of applications that only require moderate precision and throughput, but additionally benefit from the consolidation of different services into a smaller form factor.

Compared to existing solutions, CHP2 has several advantages and limitations, which we explicitly acknowledge. GPS provides high-precision positioning services over the entire planet, and GPS receivers are small and efficient. Unfortunately, broadcast systems such as GPS are particularly susceptible to spoofing and jamming cyberattacks [20,21,22]. Cooperative ranging systems, such as CHP2, cannot achieve the same positioning precision and coverage area, but they are naturally more resilient to these forms of cyberattacks, so they may be suitable for applications that care more about security than precision. Furthermore, cellular network infrastructure enables significantly faster network communications throughput than small systems such as CHP2. Integrating various sub-tasks with modern 5G and LTE infrastructure is a rich research topic in its own right, but we are more focused on the subset of problems that either do not need massive communications throughput, do not want the additional layer of complexity incurred by integrating with cellular networks, or do not have access to this infrastructure at all (natural disasters, civil conflicts, ad hoc deployments, etc.).

### 1.4. Hardware Implementation

We implemented and validated a CHP2 prototype on a COTS evaluation board, which we discuss in greater detail later in this manuscript. This implementation includes the Xilinx Zynq UltraScale+ MPSoC ZCU102 evaluation kit and an ADI FMCOMMS5 RF evaluation board. This prototype has a reasonably small form factor (∼0.01 m${}^{3}$) and moderately low weight (<1 kg), and supports a 10 MHz signal bandwidth operating at carrier frequencies of 915 MHz in the U.S. and 783 MHz in Europe. We support this hardware with a custom transmit-receive switching and amplifier board and custom printed antennae; while these additions are no longer within the scope of ”COTS”, they are not fundamental to the design, as we previously demonstrated a working prototype with simple dipole antennas and separate transmit-receive paths on the same computational hardware [12]. The details of this implementation and the processing architecture are discussed in greater detail later in this manuscript. This prototype is depicted in Figure 2.

### 1.5. Contributions

In this manuscript, we make the following contributions:Implement a modern TWR system on COTS hardware that simultaneously enables network communications, distributed coherence, and relative positioning using a single, narrowband waveform.Implement and validate efficient, real-time, oversampled ToA estimation techniques on this COTS processing hardware.Provide detailed descriptions of the processing architecture and firmware implementation on the ZCU102 evaluation hardware.Demonstrate high-precision, over-the-air ranging capabilities (<2 cm cabled, <10 cm airborne) using moderately low bandwidth (10 MHz) in experimental flight tests.

### 1.6. Organization

The remainder of this manuscript is organized as follows: in Section 2, we conduct a brief survey of the relevant literature and compare the CHP2 system to many other suitable alternatives; in Section 3, we define the underlying models used to develop the supporting estimation techniques and algorithms; in Section 4, we define the ToA and ToF estimation techniques to be implemented on the hardware platform; in Section 5, we describe the system architecture and specific parameters of the hardware prototype; in Section 6, we describe the hardware implementation of the prototype system; in Section 7, we describe the processing architecture of the hardware prototype; in Section 8, we present over-the-air experimental results for several flight tests; and in Section 9, we provide concluding remarks on the relevance and impact of the CHP2 system in the context of urban air mobility and UAV-enabled wireless networks.

## 2. Background

The CHP2 system is an APNT technology that incorporates results from several fields of study. In this section, we briefly discuss some of the most relevant publications in these fields.

### 2.1. Positioning, Navigation, and Timing (PNT) Systems

Numerous PNT systems provide various services using a variety of techniques. The GNSS is the most ubiquitous PNT service, which provides reliable positioning and timing services to a massive coverage area [23,24]. Unfortunately, broadcast systems are becoming increasingly vulnerable to spoofing cyberattacks [20,21,22], so many alternative approaches have been considered for safety-critical transport applications, such as self-driving vehicles or “flying cars” [25]. In [10], the authors discussed five common APNT technologies: DME [10,26]; P-WAM [27,28,29,30,31]; PL [32]; VOR [10]; and the LDACS1 [33,34]. In Table 1, we broadly compare the ranging precision of CHP2 to other traditional PNT systems such as the WAAS [35]; RTK GPS [36]; the UHARS [37]; and UWB systems [38,39,40]. Please note that this is a highly diverse set of technologies that are subject to numerous different design decisions and variables that may affect their performance. We do not present this information to make any claim that our proposed solution is necessarily better or worse than related systems; rather, we aim to broadly categorize similar technologies to provide a general idea of what they are capable of and to highlight some key similarities and differences.

### 2.2. Vehicular Applications

CHP2 was originally designed to provide robust PNT services to automated vehicular applications, both terrestrial and airborne; however, many emerging technologies enable similar capabilities through different RF techniques. These V2V positioning technologies include radar systems [42]; LiDAR [43,44]; optical systems [45,46]; and RFID [47]. Many RF systems have also been considered for collision avoidance, including radar systems for ground vehicles [48,49] and UAVs [50,51]; Li-Fi systems [52,53]; and LiDAR systems [54,55,56]. Other related applications include ATM [57,58] and asset tracking [59], which require cooperative positioning and communications services. In many applications, these technologies offer excellent performance but are limited in various respects, including flexibility, SWaP-C, interference, and resource consumption. We propose CHP2 not as a replacement for these systems but as a suitable alternative that offers efficiency and flexibility to adapt to future design challenges.

### 2.3. COTS RF Hardware

Despite the incredible capabilities of these emerging technologies, implementing and validating functional prototypes is still a tedious and time-consuming challenge. SDR [60,61,62] and RFSoC [18,19] have greatly simplified this process but still require some expertise to use efficiently. One of the most popular SDR platforms is the Ettus USRP series [63,64], which supports both the UHD [65,66] and GNURadio [67,68,69] to greatly simplify the development of simple radio applications. For more advanced applications, Xilinx manufactures a range of RFSoC that integrate processing systems with programmable logic and RF hardware [18]. These platforms enable more advanced features and—naturally—are more difficult to program.

### 2.4. Spectral Convergence

Spectral convergence refers to an emerging class of co-operation, co-existence, and co-design techniques that allow modern radio systems to adapt to cluttered environments and exploit their neighbors for mutual benefit [11]. Previous studies [70,71] demonstrated that co-design techniques offer better joint performance than traditional interference mitigation approaches to radar and communications coexistence. Spectral convergence promises significant performance enhancements for several RF systems, including radar systems for ground vehicles [48,49] and UAVs [50,51]; Li-Fi systems [52,53]; and LiDAR systems [54,55,56]. These approaches offer high-performance solutions but must be carefully co-designed to mitigate interference to other systems on the same platform. Other related applications include ATM [57,58] and asset tracking [59], which require cooperative positioning and communications services that greatly benefit from cooperative design strategies.

### 2.5. Distributed Coherence and MIMO Extensions

Distributed coherence refers to the synchronization and simultaneous operation of a distributed network of systems. CHP2 enables the distributed coherence implicitly by synchronizing each user and estimating the relative position of every node in the network. One of the original time synchronization algorithms is known as the NTP [72,73]. Numerous time synchronization algorithms have been deployed in modern sensor networks [74,75]. Several other studies have also discussed algorithms for maintaining distributed coherence in the context of distributed transmit beamforming [16,76,77]. In this study, we use an extended version of NTP that jointly estimates clock parameters and ToF [15]. Similar ranging and synchronization systems have been considered previously [17] but have not yet been applied to modern hardware platforms.

Multi-antenna, distributed coherent systems also enable powerful applications such as advanced relay networks [78], energy-efficient multi-beamforming [79], multiple access networks [80], and IoT connectivity [81]. The CHP2 prototype presented in this work is a multi-antenna system intended to support this class of applications, but we have not yet demonstrated these capabilities on our early hardware demonstration platform.

## 3. Model Definitions

In this section, we define the time and propagation models used to develop the CHP2 estimation and filtering techniques. Users interact by transmitting and receiving an RF waveform with bandwidth *B* at a carrier frequency $f_{c}$. We assume a standard line-of-sight propagation model [82], but use a non-standard, high-fidelity timing model to capture the subtleties of misaligned and imperfect clock sources [83].

### 3.1. Time Model

Each user operates with an independent, imperfect clock source. These clock sources are misaligned, so at any given instant *n*, there is a nominal time difference $T^{\left(n\right)}$ between any two users. These clock sources operate at the same nominal clock frequency $f_{clock,nominal}$ but can not perfectly maintain this frequency. Thus, the time offset $T^{\left(n\right)}$ is time-varying and its derivatives ${\dot{T}}^{\left(n\right)}$ and ${\ddot{T}}^{\left(n\right)}$ are assumed to be nonzero [12].

Consider two users labeled A and B, such that at time instant *n*, clock A displays a later time than clock B. Define $T^{\left(n\right)}$ as the time difference between clock B and clock A, such that if clock A displays a later time than clock B, $T^{\left(n\right)}$ is positive. This configuration is depicted in Figure 3.

User A transmits a waveform at time instant *n*, labelled $t_{A,Tx}^{\left(n\right)}$. This waveform takes a small amount of time to propagate to user B, labeled $\tau^{\left(n\right)}$. User B receives the waveform at time $t_{B,Rx}^{\left(n\right)}$. If the clocks were perfectly aligned, the receive timestamp would simply be $t_{A,Tx}^{\left(n\right)}$ + $\tau^{\left(n\right)}$. Because the clocks are misaligned, user B measures the event earlier with respect to clock B, thus the received timestamp, as perceived by user B, is [17]
(1)$$t_{B,Rx}^{\left(n\right)}=t_{A,Tx}^{\left(n\right)}+\tau^{\left(n\right)}-T^{\left(n\right)}.$$
At some later time instant n+1, user B transmits a waveform to user A. The received timestamp, as perceived by user A is [17]
(2)$$t_{A,Rx}^{\left(n+1\right)}=t_{B,Tx}^{\left(n+1\right)}+\tau^{\left(n+1\right)}+T^{\left(n+1\right)}.$$

### 3.2. Propagation Model

A signal *x* is transmitted through a line-of-sight channel, during which it is distorted before the signal *z* is received. The traditional propagation model [82] for this channel is
(3)$$z=ax\left(t-\tau \right)e^{j2\pi f_{c}\tau },$$
where *a* is the complex channel attenuation. We assume a basic line-of-sight channel attenuation [84]
(4)$$a^{2}={\left(\frac{\lambda }{4\pi d}\right)}^{2}G_{Tx}G_{Rx},$$
where $G_{Tx}$ and $G_{Rx}$ are the transmitter and receiver antenna gains, λ is the signal wavelength, and *d* is the distance between the two users.

For a transmission from A to B, the misaligned clocks and relative velocity between the platforms may induce a carrier frequency offset ζ. We extend the model in [12], which includes this carrier offset, and make the appropriate substitutions [83], using (Equation 1):(5)$$z\left(t_{B}\right)=\left|a\right|x\left(t_{B}-t_{A,Tx}^{\left(n\right)}-\tau^{\left(n\right)}+T^{\left(n\right)}\right)e^{jp}+n\left(t_{B}\right);$$
(6)$$p=2\pi \left(\tilde{\phi }+\zeta_{B}t_{B}-\left(f_{cr,n,B}+\epsilon_{B}+\zeta_{B}\right){\bar{\tau }}_{B}\right),$$
where $t_{B}$ is the clock B time reference, *n* is complex additive white Gaussian noise, $\tilde{\phi }$ is the phase noise induced by the hardware, channel, and miscalibration, $f_{cr,n,B}$ is the nominal clock B carrier frequency, $\epsilon_{B}$ is the error between this nominal frequency and the actual frequency, and ${\bar{\tau }}_{B}=\tau^{\left(n\right)}-T^{\left(n\right)}$. The corresponding propagation model from B to A may be obtained by making the relevant substitutions [83].

## 4. Estimator Definitions

The receive timestamps discussed in Section 3 are estimated using three ToA estimation techniques: (a) a critically sampled matched filter [15], (b) an oversampled reference bank [14], and (c) an interpolated correlator [83].

### 4.1. Critically Sampled Matched Filter

Consider the baseband received signal *z* and a known baseband reference signal *x*. These signals are sampled by an ADC operating at a sampling frequency $f_{s}$, such that
(7)$$z\left[m\right]=z\left(mf_{s}^{-1}\right),\;m\in \left[0,1,...,M-1\right],$$
where *m* is the sample index, and *M* is the total number of samples [15]. This sampling rate is typically 2 or 4 times the signal bandwidth for basic communications systems. These sampled signals are used to evaluate the cost function
(8)$$g\left[k\right]={\left|\sum_{m=0}^{M-1}z\left[m\right]x^{*}\left[m-k\right]\right|}^{2},\;k\in \left[0,1,...,K-1\right],$$
where ${}^{*}$ denotes the complex conjugate, *k* indexes integer time shifts, and *K* is the total number of tested time shifts [15]. This cost function compares z and x at each integer shift *k*, and is maximized for the *k* that best aligns the signals in time. The time domain ToA estimate $\hat{\nu }$ is therefore
(9)$$\hat{\nu }=\hat{k}f_{s}^{-1}\;;\;\;\hat{k}=\arg\max_{k}g\left[k\right].$$

The precision of this technique is limited to the time difference between two adjacent test points *k* [15]. For a bandwidth of 10 MHz and a sampling rate of 40 MHz, the time difference between the two samples is 25 ns, which corresponds to 7.5 m for a signal traveling at the speed of light. This technique is computationally efficient and easy to implement but does not provide the desired resolution for the system [82].

### 4.2. Oversampled Reference Bank

The fundamental resolution of the matched filter may be increased by oversampling the signals at a common multiple of the original signal bandwidth, but resampling is a computationally expensive operation and increases the number of multiplications *M* per test point and the total number of test points *K*. This quickly becomes prohibitively expensive.

We may limit this computational complexity by oversampling the reference signal *x* by a large factor (order 200), building a reference bank of shifted versions of *x* at this oversampled frequency, then downsampling the entries in this reference bank back to $f_{s}$. This bank contains copies of *x* at the original sampling frequency $f_{s}$, but each adjacent shift lies on the much more precise sampling lattice defined by the large oversampling frequency [14]. This allows us to test ToA hypotheses with much higher temporal resolution while still performing computations at the sampling frequency $f_{s}$.

We further limit the computational complexity by limiting the number of test points evaluated in the cost function. Instead of searching the same range of *k* that the matched filter evaluates, we instead search only in the immediate vicinity of the $\hat{k}$ from (Equation 9). Label a coarse test point *p* some small number of samples before $\hat{k}$, index the “micro-shifts” in the reference bank by *q*, define the total number of test points as *Q* and construct the reference bank B such that each row corresponds to a micro-shifted version of x, which builds a Q×M matrix. The micro-resolution and the narrow-window cost function may be written [14] as
(10)$$g^{\prime }\left[q\right]={\left|\sum_{m=0}^{M-1}z\left[m+p\right]x^{*}\left[m-q\right]\right|}^{2},q\in \left[0,1,...,Q-1\right].$$

Given the carefully structured reference bank B, the sum may be evaluated as a simple matrix multiplication,
(11)$$g^{\prime }\left[q\right]={\left|zB^{†}\right|}^{2}\;,$$
where ${}^{†}$ denotes the Hermitian conjugate. The time domain ToA estimate $\hat{\nu }$ then becomes
(12)$$\hat{\nu }=pf_{s}^{-1}+\hat{q}f_{s,bank}^{-1}\;;\;\;\hat{q}=\arg\max_{q}g^{\prime }\left[q\right],$$
where $f_{s,bank}$ is the oversampling rate of the filter bank. B may be evaluated offline and stored on the hardware, limiting the additional complexity of adding this second stage to the ToA estimation process [14].

### 4.3. Interpolated Correlator

The resolution of $g^{\prime }\left[q\right]$ may be further improved by applying a second-order polynomial fit to the correlation output and using this fit to interpolate between the samples near the peak. This over-resolves the intrinsic resolution of the sampling lattice defined by $f_{s,bank}$ but only significantly improves at high integrated SNR. A thorough analysis of these estimators is the subject of an adjacent study and is beyond the scope of this discussion. CHP2 is designed to operate at 20 dB SNR with 30 dB of waveform integration, at which point the interpolated correlator further improves the timing resolution by a factor of 10 [83].

### 4.4. ToF and Range Estimation

Given these high-precision ToA estimates, we can estimate both the time offset *T* and the time-of-flight τ between two CHP2 platforms. In an adjacent study [85], we explore several advanced ToF synchronization algorithms. For clarity, we provide only the simplest formulation that matches the experimental results presented later in the manuscript. For a two-way interaction A → B → A indexed by *m*, we can compute the difference
(13)$${\hat{\gamma }}_{A}^{\left(m\right)}=\left({\hat{t}}_{A,Rx}^{\left(n+1\right)}-t_{A,Tx}^{\left(n\right)}\right)-\left(t_{B,Tx}^{\left(n+1\right)}-{\hat{t}}_{B,Rx}^{\left(n\right)}\right),$$
where ${\hat{t}}_{A,Rx}^{\left(n+1\right)}$ and ${\hat{t}}_{B,Rx}^{\left(n\right)}$ are ToA estimates, and the transmit times are known and shared. The coarse ToF estimates are simply
(14)$${\hat{\tau }}^{\left(n\right)}={\hat{\tau }}^{\left(n+1\right)}=\frac{{\hat{\gamma }}_{A}^{\left(m\right)}}{2},$$
and the time offsets are then computed using (Equation 1) and (Equation 2).

After several cycles, we can iteratively refine these coarse estimates using previous cycles to estimate the first-order derivatives. We estimate and track the derivatives $\dot{\tau }$ and $\dot{T}$ across multiple cycles. Define the frame length $l_{A}$ as the time between consecutive transmissions. For the $k^{th}$ iteration after the coarse estimate, the ToF estimates become
(15)$${\hat{\tau }}^{\left(k,n\right)}=\frac{\gamma_{A}^{\left(m\right)}-{\hat{\dot{\tau }}}^{\left(k-1,n\right)}{\hat{l}}_{A}^{\left(k,n\right)}-{\hat{\dot{T}}}^{\left(k-1,n\right)}{\hat{l}}_{A}^{\left(k,n\right)}}{2},$$
(16)$${\hat{\tau }}^{\left(k,n+1\right)}=\frac{\gamma_{A}^{\left(m\right)}+{\hat{\dot{\tau }}}^{\left(k-1,n\right)}{\hat{l}}_{A}^{\left(k,n\right)}-{\hat{\dot{T}}}^{\left(k-1,n\right)}{\hat{l}}_{A}^{\left(k,n\right)}}{2},$$
and $\hat{T}$ is updated each iteration using (Equation 1) and (Equation 2) [85].

## 5. System Architecture

In this section, we outline the major components of the CHP2 system architecture.

### 5.1. Data Link Layer

Two users interact by alternately transmitting and receiving a joint waveform containing communications data and navigation reference sequences. Transmissions are scheduled every 100 ms, and receptions must be processed before the next transmission. This cycle is depicted in Figure 4.

The current CHP2 implementation uses four antennas on each platform. This number is not fundamental and can be changed arbitraily in different implementations. In this testbed, we only use one antenna to transmit the communications payload, but a full implementation will use all available antennas to enable proper MIMO capabilities. Different navigation sequences are transmitted at different times from different antenna, as depicted in Figure 5. This creates the necessary spatial diversity to translate the ranging estimates into position and orientation estimates.

### 5.2. Physical Layer

The joint waveform consists of a communications component followed by a positioning component. The communications component contains a payload, a pre-amble and mid-amble surrounding the payload, and a post-amble placed at the end of the waveform. This payload contains all of the ToF and time offset estimates, as well as a reserved space for arbitrary communications. The positioning component contains four orthogonal navigation reference sequences. The size and structure of these components are depicted in Figure 5.

The length of each component and critical waveform parameters are summarized in Table 2. The ambles are used for coarse acquisition and frequency correction, and the post-amble is used to precisely estimate the carrier frequency offset. The amble sequences are drawn from a set of Gold codes and modulated using BPSK modulation. The length of each amble is chosen based on the receiver operating characteristic (ROC) curve and the Cramér–Rao lower bound (CRLB) for frequency offset estimation. The length of the navigation sequences is chosen based on the CRLB for ToA estimation [15,17]. The communications payload is modulated using BPSK and temporally spread to support the range requirement of 10 km. The positioning sequences are pseudo-random sequences treated for delay-tolerant orthogonality and modulated using MSK. Further details are provided in [83].

## 6. COTS Hardware Components

The CHP2 system is implemented using COTS components and a custom transmit-receive switching board. This TR board was already available and was used for convenience, but could easily be replaced by various off-the-shelf alternatives or by simply using different paths for the transmit and receive antennas. A block diagram of the hardware platform is depicted in Figure 6.

### 6.1. Motherboard – ZCU102

This hardware implementation is primarily driven by a Xilinx Zynq UltraScale+ MPSoC ZCU102 evaluation kit [18], which acts as the central processor for the rest of the system. This kit features the flagship Zynq UltraScale+ MPSoC with a quad-core Arm Cortex-A53, dual-core Cortex-R5F real-time processors, and a Mali-400 MP2 graphics processing unit for programmable logic, enabling a range of processing options and capabilities [86]. We interface this central unit to an Analog Devices daughterboard to generate RF signals and leverage the numerous interfaces to log data files, collect telemetry, and monitor the resource utilization. We power this board using basic lithium polymer UAV batteries and a simple off-the-shelf power supply. This board (fully assembled) is depicted in Figure 7.

### 6.2. Transceiver—FMCOMMS5

To generate RF signals, we interface the ZCU102 with an Analog Devices FMCOMMS5 RF evaluation board [87]. This board features dual AD9361 RF Agile Transceivers and supports up to 4 × 4 MIMO configurations [88]. This board interfaces with the ZCU102 using two standard FMC connectors. This device is depicted in Figure 8.

### 6.3. Transmit-Receive Switching Board

For a related problem, we built a custom transmit-receive switching and amplifier board, which effectively combines a TR switch and power amplifiers into a nice form factor. This component is not COTS, but could be substituted with any number of standard TR switches and amplifiers without loss of generality and with only slightly worse noise figure and slightly more power consumption. The power amplifiers (PAs) provide 30 dB gain on transmit up to 30 dBm. The LNA provide 20 dB gain on receive. For testing, both of these amplifier chains can be bypassed. This board also includes an external 40 MHz OCXO clock source that can be used if necessary. This device is depicted in Figure 9.

### 6.4. Antennas

For this implementation, we used simple printed circuit board omni-directional antennas, depicted in Figure 10. In a previous iteration, we also demonstrated that rudimentary dipole antennas are also sufficient for close-range operations [12]. These antennas cover both the US 915 MHz ISM band and the EU 783 MHz licensed experimental bands used for this prototype. These antennas connect to the FMCOMMS5 board using basic SMA cables, which are fed through carbon fiber tubes for stability and mounted to the aluminum frame using plastic brackets (orange).

### 6.5. Enclosure

We mounted all of the electronics to a simple folded aluminum enclosure with cutouts to reduce weight. For the ground user, we enclosed the completed device in a Pelican case and routed cables through the case to connect to the external antennas. For the UAV user, we mounted the completed device directy to the undercarriage of our DJI S1000+. This is depicted in Figure 11. The case itself is approximately 20 cm × 20 cm × 5 cm and weighs a little less than 1 kg. This UAV already has storage trays for multiple lithium polymer batteries so we placed an additional 14.8 V, 3700 mAh battery to power the electronics next to the one powering the UAV itself, and connected it with a standard XT90 extension cable.

## 7. Processing Architecture

An overview of the processing chain is depicted in Figure 12. The processing architecture follows a typical communications OSI model. This includes an RF layer, physical layer, data link layer, and application layers. The system functions are organized by layer in Figure 13.

### 7.1. CHP2 Physical Layer

The CHP2 physical layer is distributed across the processing system (PS) and programmable logic (PL) components of the MPSoC, as depicted in Figure 13 and Figure 14. The skeleton is based on the Analog Device SDR platform [89]. Timing-critical components are implemented as HDL IP cores in the PL section. The Tx and Rx engines are customized and placed on the IQ data paths. The correlator reference bank, the TRS board controller, and the carrier-synthesizer phase-reset controller are placed on the AXI bus. In the PS section, CHP2 processes are run on the ARM Cortex-A53 cores. This includes waveform encoding and decoding, channel equalization, frequency offset estimation and correction, and automatic gain control.

#### 7.1.1. Transmit Engine

The transmit engine (Figure 15) is a large IQ data buffer collaborating with a timer-controlled finite state machine. This engine is driven by the 40 MHz OCXO on the TRS board. The state machine periodically designates a transmit timestamp for the CHP2 processes. The data buffer is populated with raw IQ data through the AXI bus. Once the transmission time is reached, the buffer content is transferred to the AD9361 control core. The IQ data is loaded into local memory using the AXI DMAC, which resolves compatibility issues with existing Analog Devices drivers.

#### 7.1.2. Receive Engine

The receive engine (Figure 16) captures received signals and transfers the data to the PS section as in Figure 14. During the acquisition stage, a frame detector continuously correlates the received data with the waveform pre-amble. If this correlation exceeds a pre-defined threshold, the detector declares a detection and dumps the raw IQ data into the DMA blocks, which then move to the PS section. The frame detector saves the timestamp associated with this event, which corresponds to the matched filter ToA estimate $\hat{k}$, as described in (Equation 9).

#### 7.1.3. Reference Bank ToA Estimation

The reference bank ToA estimator defined in Section 4.2 is implemented using a computation-efficient structure. This estimator performs a complex matrix multiplication, searches for the maximum in the resulting vector, and reports the corresponding index. This method is efficient but requires loading a large bank of coefficients into the block memory of the programmable logic, which is unrealistic.

To reduce the number of coefficients that must be loaded into memory, we exploit the structure of the matrix and save only a subset of the rows. For an oversampling rate of 2 GHz and a sampling rate of 40 MHz, 50 micro-shifts correspond to 1 sample shift, meaning that every subsequent set of 50 rows is identical to the first 50 shifted by some integer number of samples. Instead of performing the full matrix multiplication, we can test only 50 rows against several sample-shifted versions of the received signal to emulate the output of the full multiplication without any loss. The structure of this reference bank is depicted in Figure 17.

The polynomial fit described in Section 4.3 may be applied to improve the precision of the reference bank correlator. We extract 7 samples before and after the reported maximum and build a least-squares estimator for the coefficients of a second-order polynomial. Label the indices of these samples $x_{-7},x_{-6},...,x_{6},x_{7}$ and the corresponding correlator outputs $y_{-7},y_{-6},...,y_{6},y_{7}$. These 15 test points are used to estimate the polynomial fit to
(17)$$\begin{array}{cc} y & =\beta_{0}+\beta_{1}x+\beta_{2}x^{2}+\epsilon . \end{array}$$
The corresponding least-squares estimator takes the form
(18)$$\begin{array}{cc} \left[\begin{array}{c} y_{-7}\\ \vdots \;\\ y_{6}\\ y_{7} \end{array}\right] & =\left[\begin{array}{ccc} 1 & x_{-7} & x_{-7}^{2}\\ \vdots & \vdots & \vdots \\ 1 & x_{6} & x_{6}^{2}\\ 1 & x_{7} & x_{7}^{2} \end{array}\right]\left[\begin{array}{c} \beta_{0}\\ \beta_{1}\\ \beta_{2} \end{array}\right]\;, \end{array}$$
(19)$$\begin{array}{cc} \hat{\beta } & ={\left(X^{†}X\right)}^{-1}X^{†}y\;, \end{array}$$
where y, X, and β are the matrixes defined in (Equation 18). For a 4 × 4 MIMO system with 16 transmit-receive pairs, upon reception of the joint waveform, a receiver estimates the ToA of all navigation sequences on all receive channels in parallel. The hardware implementation of these 16 correlators is depicted in Figure 18.

### 7.2. Physical Layer Operations

The CHP2 physical layer also includes the following operations:Carrier Frequency Correction: CHP2 performs two carrier frequency offset (CFO) corrections on the received data. The first correction coarsely estimates the CFO using the pre- and mid-ambles and applies the correction to the data being passed to the communications decoding processing chain. The second correction finely estimates the CFO using the pre- and post-ambles and applies the correction to the data being passed to the ToA correlation engine.Pulse Shaping Filter: We apply a raised cosine FIR filter in the Tx and Rx engines to limit the spectral leakage. This filter uses 65 taps, a 0.25 roll-off factor, a 16 symbol span, and 4 samples per symbol.Channel Equalizer: We apply a 5-tap adaptive Wiener filter to the received data in the communications processing chain to mitigate multi-path effects. This filter is evaluated on each reception using the pre-amble.Automatic Gain Control: We employ automatic gain control (AGC) to maintain a received instantaneous SNR between 20 and 25 dB. This prevents power saturation and starvation as the platforms move closer or farther apart, respectively.

### 7.3. CHP2 Data Link Layer

The data link layer schedules operations, including transmission and reception, frame extraction, and timestamp packaging. This scheduler also controls the TRS amplifier board. The DLL also contains an Ethernet logger process, which exports system logging data at a rate of 10 Hz. This logging data includes range, position, and orientation estimates; channel SNR estimates; timing exchange information; and hardware debugging information.

### 7.4. CHP2 Application Layer

The application layer uses the ToA estimates and timing information shared by the lower layers to drive a ToF synchronization algorithm. This algorithm estimates the distance between each of the 16 transmit-receive antenna pairs and digitally synchronizes the user clocks. The first iteration of this algorithm is described in detail in [12]; the current version is described in great detail in a companion manuscript but is beyond the scope of this discussion. The application layer also contains several Kalman filtering techniques, manual and automatic calibration processes, and position and orientation estimation techniques.

### 7.5. ZynqMP Hardware Utilization

The CHP2 system is implemented and evaluated by the Xilinx Vivado EDA tool. Programmable resource consumption is within the XCZU9EG specifications [86], and the implementation obeys the system timing constraints. About 63% of the LUT resources and 54% of the DSP blocks are utilized. The resource utilization profile is summarized in Table 3.

## 8. Experimental Results

We demonstrate the CHP2 prototype ranging capabilities using both cabled and over-the-air experiments. This implementation includes several ToA and ToF methods; the advanced estimators (cabled) demonstrate the system’s best ranging performance while the simple estimators (over-the-air) are more robust and suitable for a field demonstration. This initial implementation uses a fixed-rate protocol, so the communications rate is not variable and we limit the operating range (10 km) to comfortably ensure that the bit error rate at the output of the decoder is 0. We explore the manifold optimization of data rate, ranging precision, and timing synchronization and the use of adaptive rate protocols in companion studies. The objective of the following experiments is to demonstrate the successful implementation of the proposed system on a COTS experimental testbed and to provide the initial validation of the previously-claimed 10 cm ranging resolution.

### 8.1. Cabled Experiments

We performed cabled experiments to demonstrate the achievable ranging precision of the ToF estimation algorithm and Kalman filtering techniques. Two 4-antenna CHP2 users are connected by RF cables, attenuators, and a signal combiner to emulate an ideal LoS propagation channel. We subtract the mean of the resulting ToF estimates to highlight the variance and stability. These normalized ToF estimates are presented in Figure 19 for the interpolated correlator bank estimator and two extended Kalman filter formulations. The unfiltered correlation results maintain a ranging precision below 1.7 cm, while the extended Kalman filter(EKFs) drive the ranging precision as low as 0.3 cm at the cost of increased settling time.

### 8.2. Over-the-Air Demonstrations

We performed over-the-air flight demonstrations at the general aviation airport in Nördlingen, Germany. We assembled a 4-antenna ground station using telescoping antenna mounts and a UAV user by mounting the hardware enclosure to a DJI S1000+, depicted in Figure 20. The following experiments were performed at a carrier frequency of 783 MHz using 10 MHz bandwidth. This demonstration only uses the simple ToF estimation technique defined in Section 4, not the Kalman filter included in the cabled results.

#### 8.2.1. VTOL Test

To demonstrate over-the-air ranging capabilities, we executed a simple VTOL manuever and plotted the range estimates against a tachymeter reference in Figure 21. The antennas and tachymeter are physically separated on the UAV, which induces a small range difference when plotted together. The system maintains accurate range estimates during all stages of the maneuver, even when the tachymeter reference fails at 90 s. During take-off and landing, performance is worsened by ground-bounce multipath. At the apex, the drone cannot maintain a perfect hover due to strong winds, so the range estimates appear artificially noisy, since the tachymeter reference is mounted to the center of the platform, and the antennas are mounted at the corners.

The achieved ranging precision is summarized in Table 4 for the duration of the maneuver. Most channels maintain a standard deviation of around 10 cm except for UAV channel 1, which was obscured by the UAV frame. This precision may be increased by implementing the interpolation technique demonstrated in the previous result, which was developed after this experiment to improve performance. The ranging performance in this configuration is also limited by ground-bounce multi-path.

#### 8.2.2. Longer-Range Tests

To demonstrate that this system is not limited to extremely close ranges, we demonstrate ranging stability in two longer-range demonstrations. During the first test, the UAV flies 200 m down the runway, lands, an operator physically interacts with the drone, and the UAV flies back to the starting position. During the second test, the UAV flies 700 m down the runway, turns around, and returns to the starting position. The results of these demonstrations are depicted in Figure 22.

## 9. Conclusions

We designed, implemented, and validated the CHP2 system to simultaneously enable communications, synchronization, and ranging from a single radio platform. In this manuscript, we described (a) the architecture of this proposed system; (b) the estimation techniques that enable it; (c) our choice of consumer-grade hardware on which to implement it; (d) a comprehensive summary of the hardware processing architecture; and (e) some simple experimental validation results. Using this initial COTS hardware prototype, we demonstrated a ranging precision below 10 cm in over-the-air flight tests and as low as 3 mm in a controlled laboratory environment. Because these ranges are generated using a time-of-flight estimation algorithm, the ranging subtask intrinsically enables time synchronization. In this early iteration, the communications protocol operates at a fixed data rate, which has already been extended to adaptive-rate protocols in parallel investigations.

Using this initial hardware implementation and concept validation, we are already exploring the following developments: (a) formulating and solving a manifold optimization problem to trade performance between the ranging, synchronization, and communications subtasks; (b) formulating more sophisticated time-of-flight estimation algorithms that generate higher ranging and synchronization precision using a variety of optimal one-shot estimators and adaptive Kalman filtering techniques; (c) implementing adaptive-rate protocols to enable better, on-the-fly flexibility in the communications data rate without modifying the firmware; (d) enabling modern distributed MIMO applications such as distributed beamforming and IoT using networks of multi-antenna CHP2 platforms; and (e) implementing traditional and reinforcement learning resource management techniques to enable and manage network applications using distributed CHP2 platforms.

## Figures and Tables

**Figure 1 sensors-23-01343-f001:**
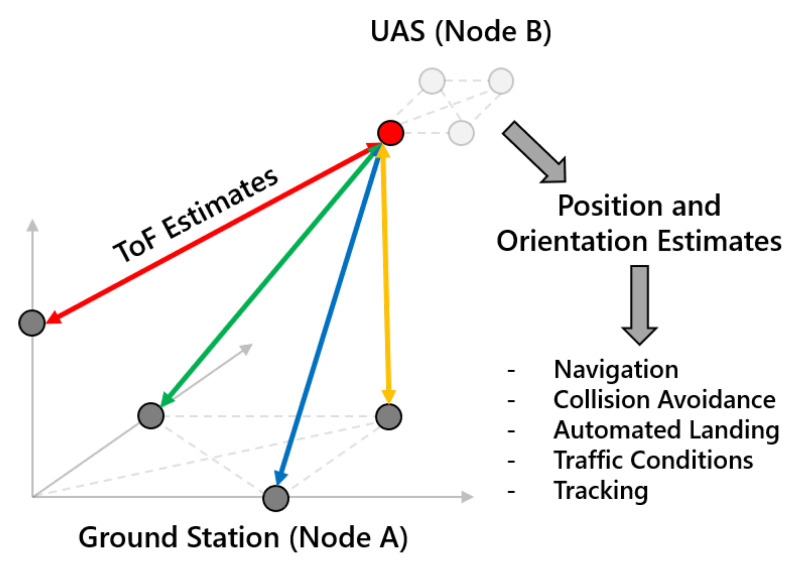
CHP2 is flexible and reconfigurable, which enables diverse network configurations on a variety of platforms. CHP2 supports both ground and aerial users, and can be tuned for both close-range and long-range applications. G2G links enable applications such as traffic management, knowledge distribution, and automatic navigation. A2G links enable applications such as ATM; flexible traffic surveillance; PNT; automated landing; and CNS. A2A links enable airborne applications such as collision avoidance, formation control, PNT, and CNS without relying on satellite links (which are susceptible to spoofing cyberattacks) or ground links—which are not always available.

**Figure 2 sensors-23-01343-f002:**
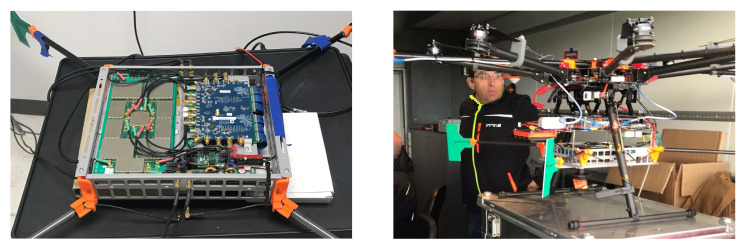
Initial hardware prototype of the CHP2 system: (**left**) the ZCU102 evaluation kit and FMCOMMS5 RF evaluation board installed in a folded sheet metal case, with dimensions 20 cm × 20 cm × 5 cm, and (**right**) the CHP2 assembly mounted to the DJI S1000+ UAV for flight testing.

**Figure 3 sensors-23-01343-f003:**
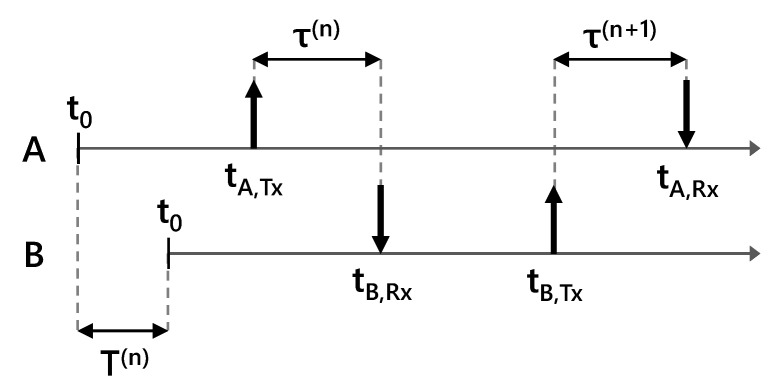
Depiction of two interactions between radios A and B. Interactions are indexed by *n*. The time offset is labelled $T^{\left(\cdot \right)}$ and the propagation delay is labelled $\tau^{\left(\cdot \right)}$. Transmit events are labeled with up arrows, and corresponding receive events are labeled with down arrows. The clocks are misaligned, so the receive timestamps $t_{(\cdot ),Rx}^{\left(\cdot \right)}$ follow (Equation 1) and (Equation 2).

**Figure 4 sensors-23-01343-f004:**
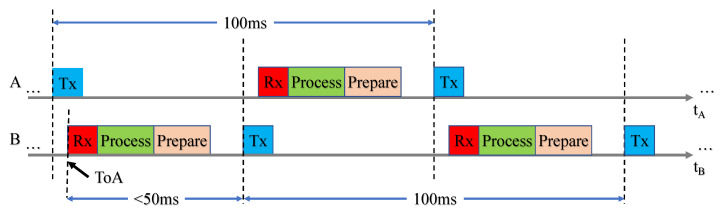
Depiction of the data link layer for two users labeled *A* and *B*. These users alternate transmitting and receiving every 50 ms. To ensure that the appropriate timing information is shared in the next frame, each user must finish executing their processing chain before the next transmission.

**Figure 5 sensors-23-01343-f005:**
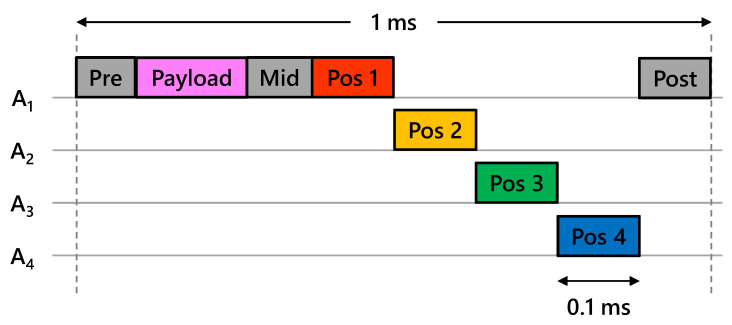
Depiction of the transmissions sent from each antenna for a single transmit frame. The current implementation only uses one antenna for the communications tasks but future implementations will use all available antennas to enable MIMO capabilities. The navigation sequences are transmitted from each antenna following a simple tim-division strategy.

**Figure 6 sensors-23-01343-f006:**
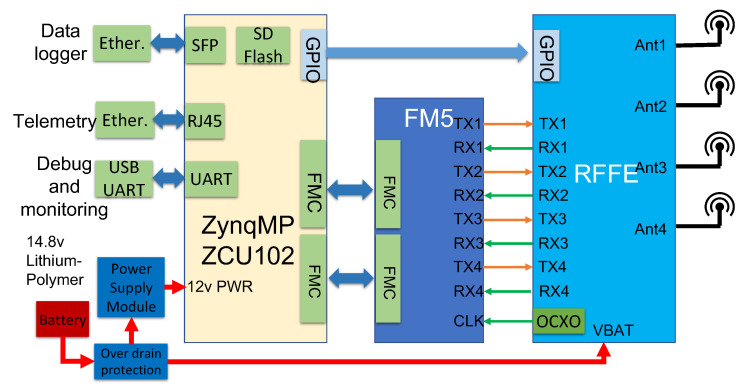
Block diagram of the hardware acrhitecture of the CHP2 prototype implementation. The system is driven by a ZynqMP ZCU102 evaluation kit, which interfaces with and ADI FMCOMMS5 RF evaluation board to produce RF signals, a transmit-receive switching and amplifier board to manage transmit and receive operations, and supporting hardware such as power supplies, data interfaces, and antennas.

**Figure 7 sensors-23-01343-f007:**
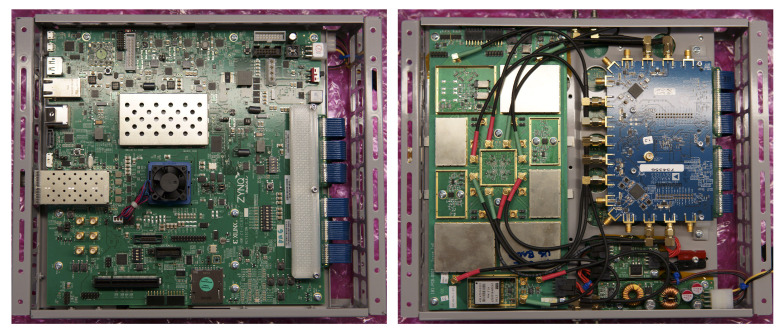
Front (**left**) and back (**right**) view of all hardware mounted in an aluminum frame. The front is the Xilinx ZCU102 evaluation board. The back contains the ADI transceiver card, the TR switching board, and the power supply module.

**Figure 8 sensors-23-01343-f008:**
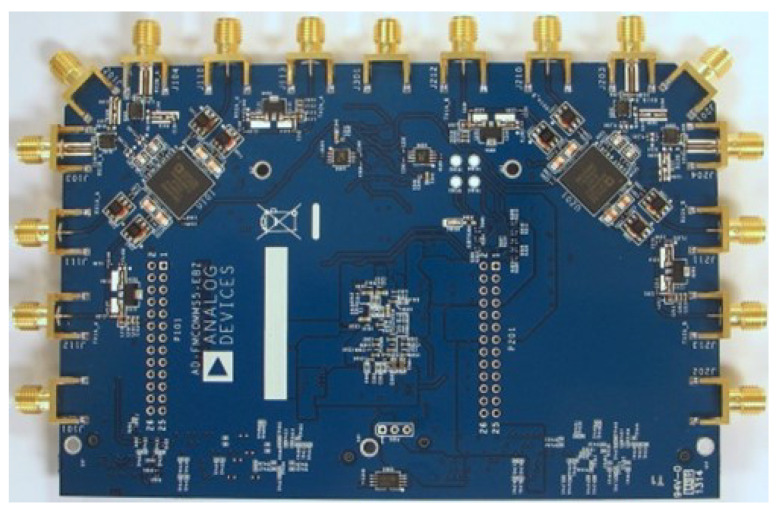
Analog Devices FMCOMMS5 RF evaluation board [87].

**Figure 9 sensors-23-01343-f009:**
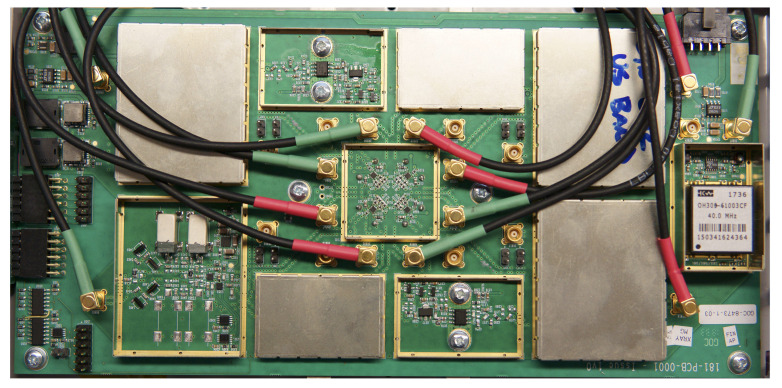
Customized transmit-receive switching and amplifier board.

**Figure 10 sensors-23-01343-f010:**
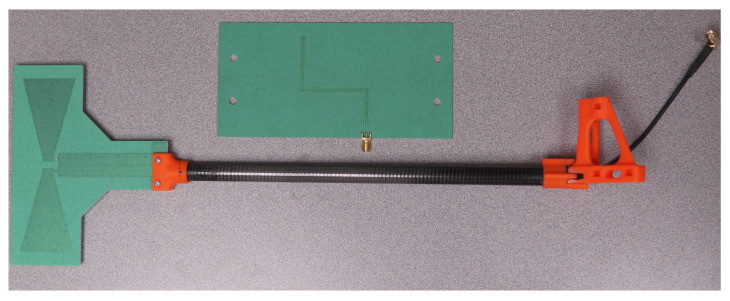
PCB omni-directional antenna, carbon fiber mounting rod, and plastic mounting brackets.

**Figure 11 sensors-23-01343-f011:**
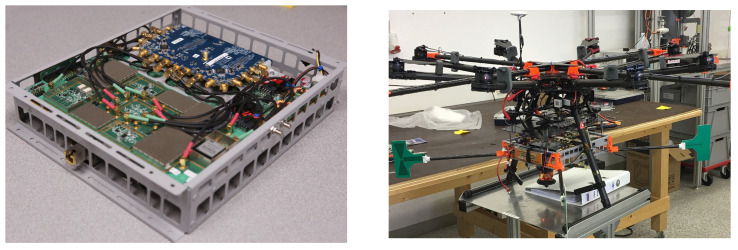
Electronics mounted in an aluminum enclosure (**left**) then mounted to the DJI S1000+ UAV platform for flight testing. (**right**) Initial hardware prototype of the CHP2 system.

**Figure 12 sensors-23-01343-f012:**
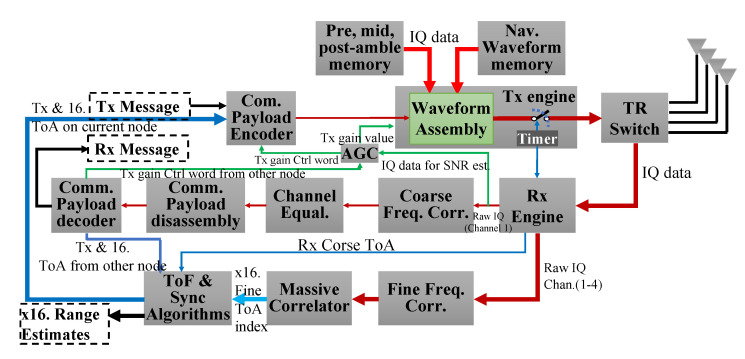
CHP2 processing chain block diagram.

**Figure 13 sensors-23-01343-f013:**
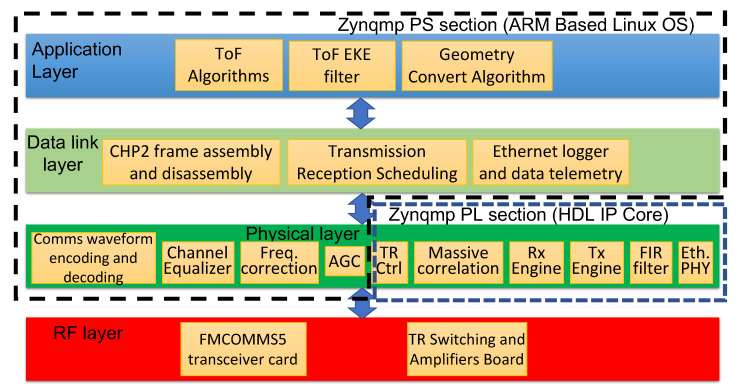
CHP2 system functions organized by OSI layer.

**Figure 14 sensors-23-01343-f014:**
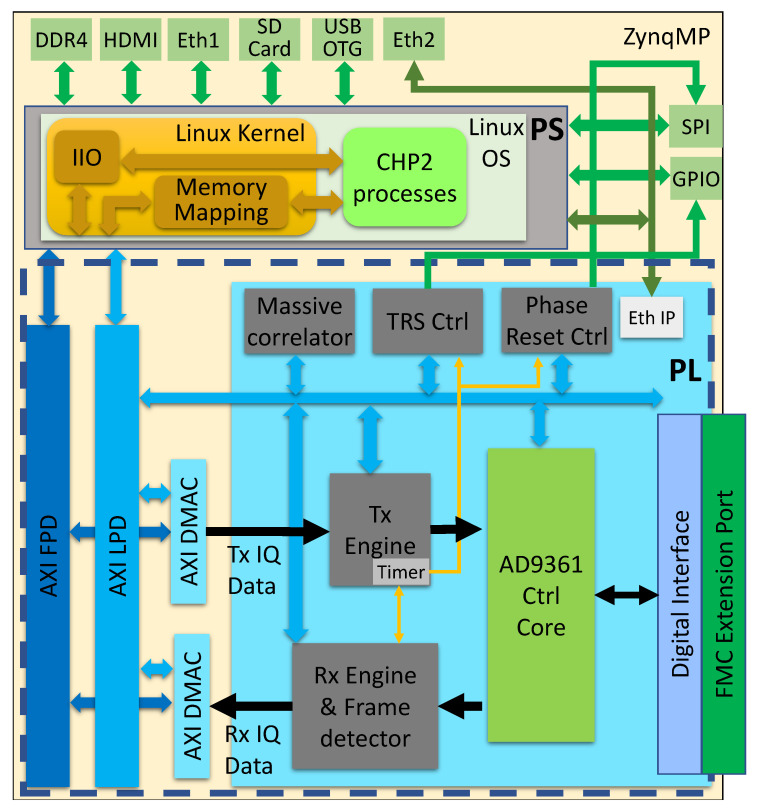
CHP2 processing architecture on the ZynqMP platform. In the PL section, the critical timing components and massive mathematical tasks are implemented as HDL IP cores. In the PS section, the light tasks are implemented as processing programs running over the ARM Cortex-A53 cores. Except for the AD9361 control core and DMAC blocks that come from the ADI SDR architecture, the remaining blocks are fully customized.

**Figure 15 sensors-23-01343-f015:**
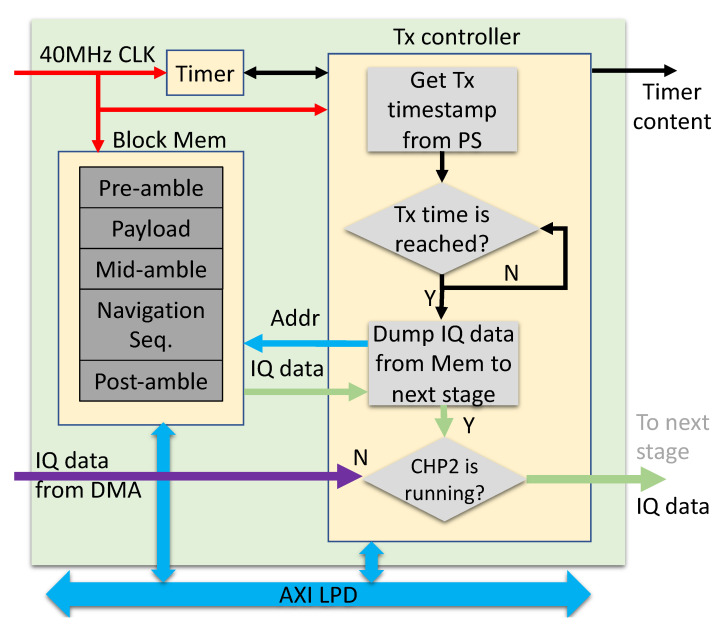
Transmit engine block diagram.

**Figure 16 sensors-23-01343-f016:**
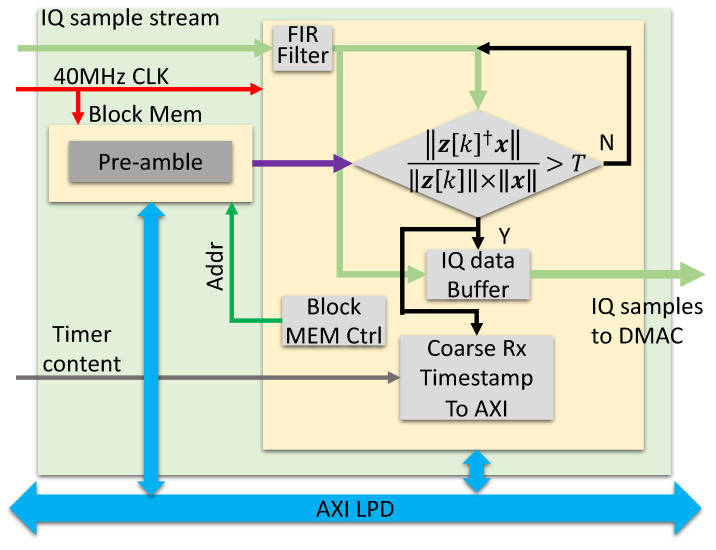
Receive engine block diagram.

**Figure 17 sensors-23-01343-f017:**
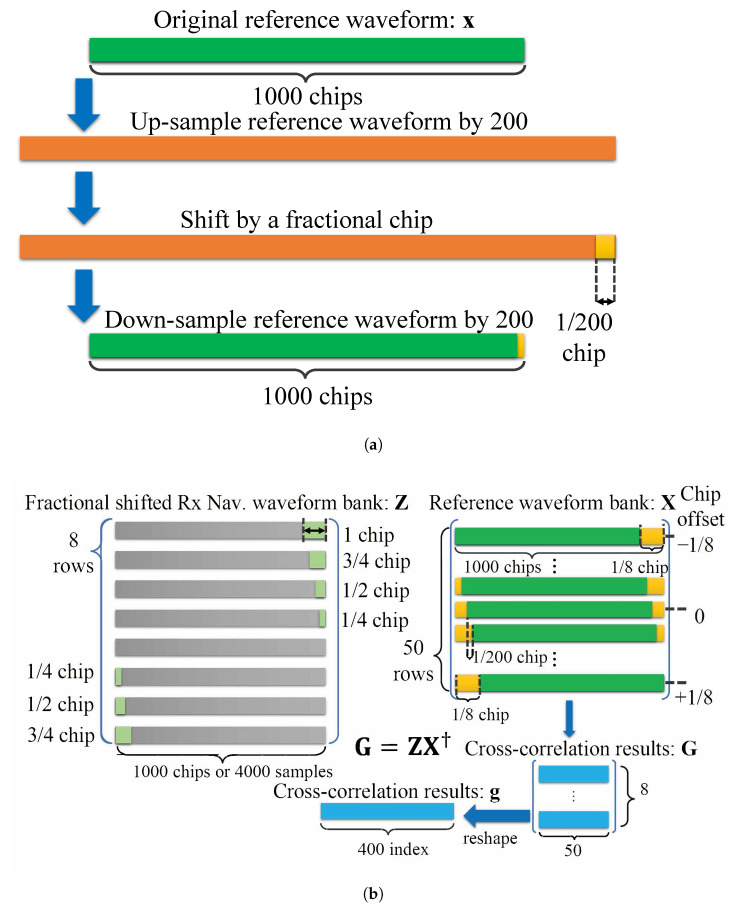
Optimized structure of the fine ToA estimation implementation. (**a**) Construct a micro-shifted version of the reference waveform. The consecutive waveforms are shifted by 1/200th of a chip. (**b**) Memory-efficient subdivision of correlator bank.

**Figure 18 sensors-23-01343-f018:**
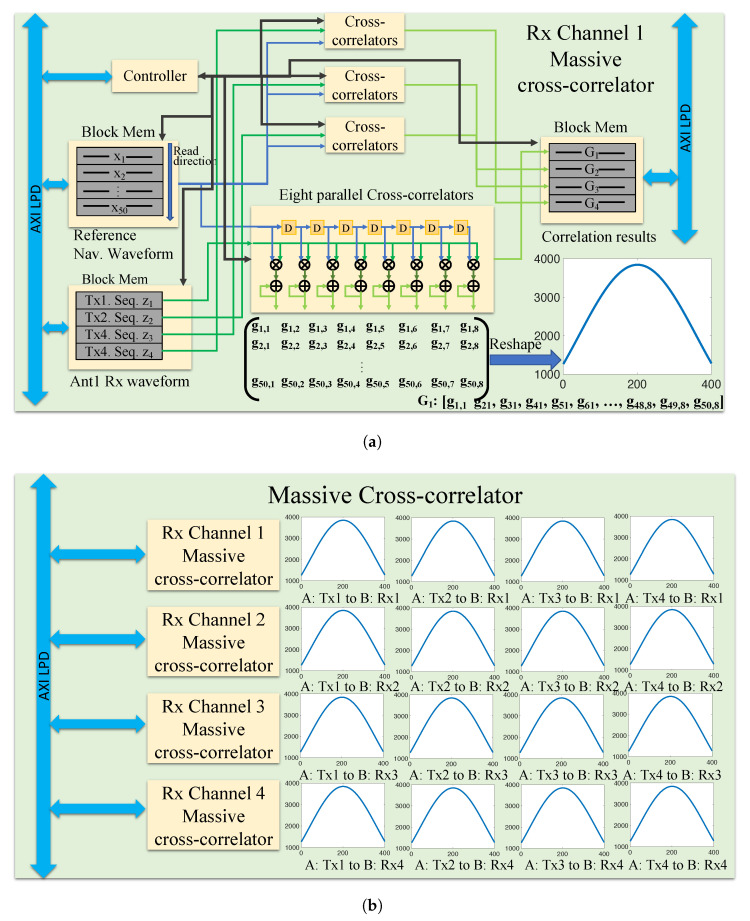
Implementation diagram of 16 simultaneous correlation banks for a 4 × 4 configuration. (**a**) Each receiving chain has four dedicated correlators. (**b**) There are four receiver correlation channels for a total of 16 correlations – corresponding to the 16 links formed by our 4-antenna units.

**Figure 19 sensors-23-01343-f019:**
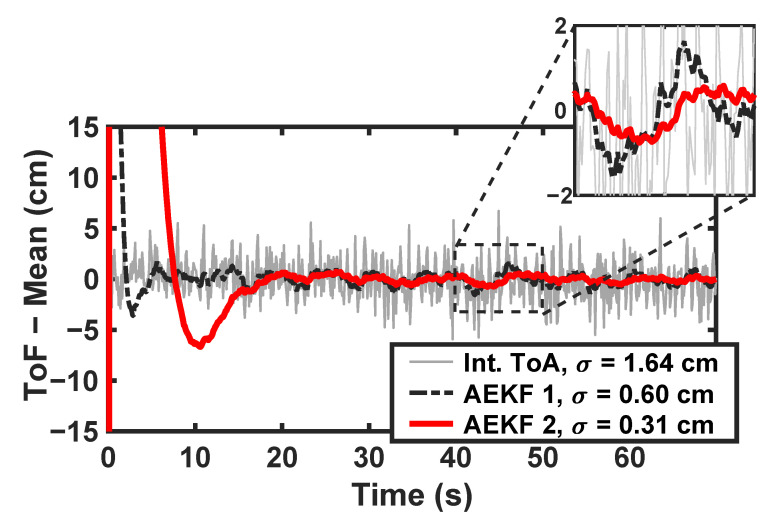
Ranging precision for 1 of the 16 antenna pairs in a cabled configuration using three different ToF estimation techniques. To highlight the estimator variance, we have subtracted the mean for clarity. The interpolated correlator estimator (gray) achieves a ranging precision (standard deviation) below 1.7 cm. The adaptive extended Kalman filter techniques (black, red) drive this precision (standard deviation) as low as 0.3 cm but require more settling time.

**Figure 20 sensors-23-01343-f020:**
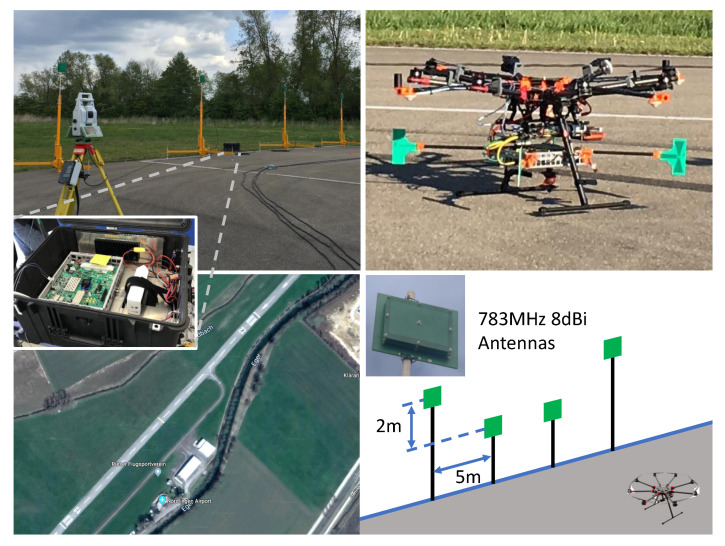
Flight test experimental configuration. One user consists of 4 patch antennas mounted on telescoping mounts (**top left**) and the hardware enclosed in a ruggedized case (**left**). The second user is mounted to a DJI S1000+ UAV for flight testing (**top right**). These experiments were performed on a runway about 700 m long (**bottom left**) using a simple antenna geometry (**bottom right**).

**Figure 21 sensors-23-01343-f021:**
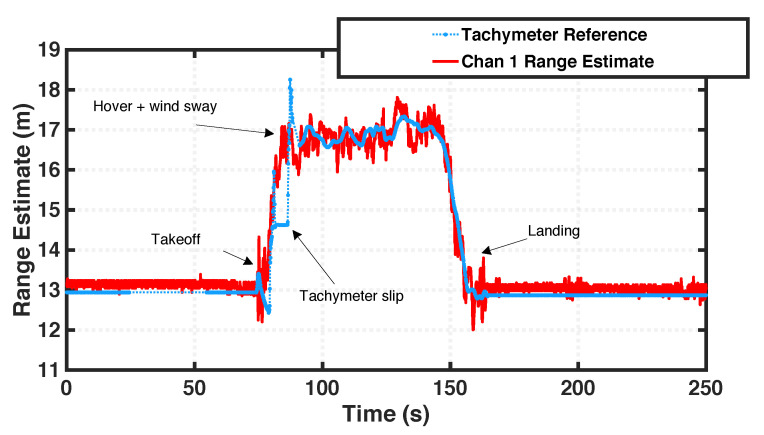
CHP2 range estimates (red) for a single transmit-receive antenna pair versus a tachymeter reference (blue) during a simple VTOL maneuver.

**Figure 22 sensors-23-01343-f022:**
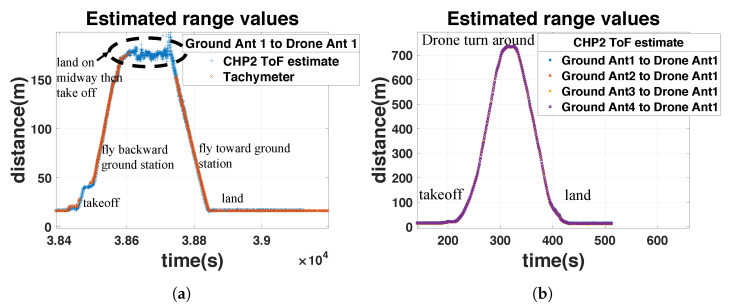
CHP2 ranging estimates for the short- and mid-range tests versus the same tachymeter reference. System performance degrades during the short-range test (left) when line-of-sight (LoS) is obstructed by the operator. The AGC maintains system stability throughout the rest of the experiments. The mid-range test is plotted without a tachymeter reference due to its range limitation (200 m). (**a**) Short-range test. (**b**) Mid-range test.

**Table 1 sensors-23-01343-t001:** Performance Comparison of CHP2, PNT, and APNT Technologies [10,26,27,28,29,30,31,32,33,34,35,36,37,38,39,40,41].

Label	Tech	Carrier	Bandwidth	Precision	Coverage
GPS (PL)	Pseudo-	1.58 GHz (L1)	1 MHz (Civ)	5–100 cm	Global
	Range	1.23 GHz (L2)	10 MHz (Mil)	1–5 cm (PL)	
GPS (WAAS)	PR		15.345 MHz	1 m	Global
GPS (RTK)	PR		15.345 MHz	1 cm	Global
LDACS1			500 kHz	20 m	370 km
VOR	AoA	108–118 MHz	10 MHz	0.35–1.4${}^{\circ }$	100–300 km
DME	RToF	960–1215	252 × 1	(Legacy)	50 km (Close)
		MHz	MHz	300 m	75 km (Mid)
				(Modern)	250 km (Long)
				100 m	
(ADS-B)	TDoA	(ADS-B)	(ADS-B)	50–100 m	(ADS-B)
P-WAM		1090 MHz	50 kHz		250 km
		978 MHz	1.3 MHz		
UHARS			10.23 MHz	20 cm	50 km
UWB			1 GHz	1–10 cm	Varied.
CHP2	Pseudo-	915 MHz (US)	10 MHz	0.1–10 cm	10 km
	Range	783 MHz (EU)			

**Table 2 sensors-23-01343-t002:** CHP2 prototype waveform parameters.

Parameter	Value	Units
Amble Length	128	chips
Payload Length	8192	chips
Nav Sequence Length	1000	chips
Bandwidth	10	MHz
Sampling Rate	40	MHz
Carrier Frequency (US)	915	MHz
Carrier Frequency (EU)	783	MHz
Transmit Power	<30	dBm
Amble Code	Gold Code, ($x^{7}+x+1$)	–
Spread Code	[1,−1,−1,−1,1,−1,−1,1]	–
Pulse Shaping Filter	Raised Cosine, β=0.25, N=16	–
Amble Modulation	BPSK	–
Payload Modulation	BPSK	–
Navigation Modulation	MSK	–

**Table 3 sensors-23-01343-t003:** CHP2 logical resource utilization on ZynqMP XCZU9EG.

Resource	Utilization	Available	%
LUT	171,436	274,080	62.55
LUTRAM	2967	144,000	02.06
FF	184,477	548,160	33.65
BRAM	402	912	44.08
DSP	1352	2520	53.65
IO	143	328	43.60
GT	1	24	04.17
BUFG	35	404	08.66

**Table 4 sensors-23-01343-t004:** Average ranging performance for all antenna pairs in the VTOL test.

Unit: centimeter (cm)				
**Antenna Pair**	**Ground** **Ant1**	**Ground** **Ant2**	**Ground** **Ant3**	**Ground** **Ant4**
Drone Ant1	16.59	14.12	11.62	11.24
Drone Ant2	13.54	8.58	7.96	8.78
Drone Ant3	11.39	8.91	8.43	9.11
Drone Ant4	10.68	9.18	9.02	10.18

## Data Availability

The data presented in this study are available on request from the corresponding author. The data are not publicly available due to the policies of Arizona State University regarding pending patents and publications.

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
