# Peer review of "Communications and High-Precision Positioning (CHP2): Hardware Architecture, Implementation, and Validation"

_sensors, 2023, doi:10.3390/s23031343_

Round 1

Author Response

For your convenience, we have prepared itemized responses to your suggestions in the attached letter.

Reviewer 2 Report

The paper contains a clear description of communication and high positioning system. For that, authors provided a little experimental campaign to validate a performance of prototype developed.

The SOA is more accurate and the section 2 of manuscript is clear and complete, but the introduction is short without references support. I suggest to add most significative references, used in section 2, to reinforce introduction section.

The section 3 and 4 are clear, but there are not references to support the mathematical treatment. In this manner, the novelty of work can be distiguish from litterature standard processing.

The sections 5 and 6 are more short, in particular the HW section is composed only by figures. 

The experimetal results are not clear, authors proposed a high-precision positioning system but there are not a table to evaluate the system proposed performance.

The conclusion is too short. The comment of results and future development are not accurately discussed.

Author Response

(The authors gave the same response as above.)

Round 2

Reviewer 2 Report

The authors followed the suggestions of the reviewers, the work is clearer and more complete.